# Clinical features and outcome of *Streptococcus agalactiae* bone and joint infections over a 6-year period in a French university hospital

Paul Loubet[1]*, Yatrika Koumar[2], Catherine Lechiche[2], Nicolas Cellier[3], Sophie Schuldiner[4], Pascal Kouyoumdjian[3], Jean-Philippe Lavigne[5], Albert Sotto[1]

1 Department of Infectious and Tropical Disease, VBMI, INSERM U1407, CHU Nîmes, Univ Montpellier, Nîmes, France, 2 Department of Infectious and Tropical Disease, CHU Nîmes, Univ Montpellier, Nîmes, France, 3 Department of Orthopedic and Trauma Surgery, CHU Nîmes, Univ Montpellier, Nîmes, France, 4 Department of Metabolic and Endocrine Disease, VBMI, INSERM U1407, CHU Nîmes, Univ Montpellier, Nîmes, France, 5 Department of Microbiology and Hospital Hygiene, VBMI, INSERM U1407, CHU Nîmes, Univ Montpellier, Nîmes, France

* paul.loubet@chu-nimes.fr

**Data Availability Statement:** All relevant data are within the manuscript and its Supporting Information files.

## Abstract

### Background

Bone and joint infections (BJIs) due to *Streptococcus agalactiae* are rare but has been described to increase in the past few years. The objective of this study was to describe clinical features and outcomes of cases of *S.* BJIs.

### Methods

We conducted a retrospective analysis of adult cases of *S. agalactiae* BJIs that occurred between January 2009 and June 2015 in a French university hospital. The treatment success was assessed until 24 months after the end of antibiotic treatment.

### Results

Among the 26 patients included, 20 (77%) were male, mean age was 62 years ± 13 and mean Charlson comorbidity index score was 4.9 ± 3.2. Diabetes mellitus was the most common comorbidity (n = 14, 54%). Six had PJI (Prosthetic Joint Infections), five osteosynthesis-associated infections, 11 osteomyelitis and four native septic arthritis.

Eleven patients had a delayed or late infection: six with a prosthetic joint infection and five with an internal fixation device infection. Sixteen patients (62%) had a polymicrobial BJI, most commonly with Gram-positive cocci (75%) notably *Staphylococcus aureus* (44%). Polymicrobial infections were more frequently found in foot infections (90% vs 44%, *p* = 0.0184). During the two-year follow-up, three patients died (3/25, 12%) and seven (7/25, 28%) had treatment failure.

### Conclusion

Diabetes mellitus was the most common comorbidity. We observed an heterogenous management and a high rate of relapse.

1 / 9

**Funding:** The authors received no specific funding for this work.

**Competing interests:** The authors have declared that no competing interests exist.

## Introduction

Group B *Streptococcus* (GBS) or *Streptococcus agalactiae* is a well-characterised pathogen of infants and pregnant women. However, invasive GBS infections are increasingly observed in non-pregnant adults (two-thirds of patients) and have become a major health concern [1]. Common clinical manifestations in non-pregnant adults include skin or soft-tissue infections, urinary tract infections, pneumonia, bacteraemia with no identified focus, arthritis and osteomyelitis [1]. GBS septic arthritis in a diabetic patient was first reported in 1940 [2], ever since the burden of GBS in invasive infections such as bone and joint infections (BJIs) is increasing [3–6]. However, few studies have described arthritis and osteomyelitis infections due to GBS in non-pregnant adults. The objective of this study was to describe clinical characteristics and outcomes of all cases of *S. agalactiae* BJIs that occurred in our hospital between 2009 and 2015.

## Materials and methods

### Study population

We retrospectively reviewed all cases of *S. agalactiae* BJIs in Nîmes University Hospital in the South of France. Our hospital has a 1,979-bed capacity, which includes one orthopaedic surgery department and one infectious diseases unit. All *S. agalactiae* BJIs including arthritis, osteomyelitis, internal fixation device infections and prosthetic joint infections were identified from the Microbiology Laboratory database using different codes: "Streptococcus"; "bone samples"; "deep samples"; "orthopaedist surgery"; "osteoarticular infection"; "bone infection". Data were reviewed from January 2009 to June 2015. Those with non-bone and joint infection were excluded.

*S. agalactiae* BJIs were identified based on past medical history with clinical evidence of infection using biological and/or radiological data, with at least two positive cultures of *S. agalactiae* identified from deep samples taken during surgery, joint aspirate samples and blood cultures. Prosthetic joint infections were defined according to the IDSA guidelines criteria [7] and classified according to the time from prosthesis implantation to the onset of infection as: early (<3 months), delayed ([3 months–2 years]) and late (>2 years) infections. Internal fixation device infections were classified according to the time of onset after implantation: early (< 3 weeks), delayed ([3–10 weeks]), and late (>10 weeks) [8].

### Data collection

All patient files were reviewed between June and September 2017 to collect the following characteristics: sociodemographic, pregnancy and comorbidities (heart failure, chronic liver disease, diabetes mellitus, inflammatory rheumatism, obesity, immunodeficiency, peripheral arterial obstructive disease, and peripheral neuropathy) as well as consumption of alcohol and tobacco. The Charlson comorbidity index (a combined age-comorbidity score used to estimate relative risk of death from prognostic clinical covariates) was calculated for each patient [9].

The location of infection and the presence of prosthetic joint or internal fixation device were recorded. The antimicrobial and/or surgical treatments performed as well as treatment outcomes, assessed at months 3, 6, 12 and 24 after the end of antibiotic treatment, were reviewed for every patient. Outcomes were assessed through phone calls and medical chart.

Treatment failure was defined as either (i) the recurrence of the same prosthetic joint infections by GBS at any time after the first line of medical and surgical treatment (i.e., relapse of infection); (ii) the recurrence of the same prosthetic joint infections because of the presence of a same species with a different antibiogram or another species at any time after the first line of medical and surgical treatment (i.e., reinfection), or (iii) death directly caused by sepsis resulting from active BJI without another known infection.

### Specimen collection and microbiological analysis

Deep biopsies samples were obtained during surgery or at patient bedside. Joint fluids, tissue samples, or bone biopsies were crushed, plated on different agar media (Columbia blood agar with 5% sheep blood agar, chocolate, Mueller–Hinton, trypticase soy, MacConkey agar plates and Schaedler tube (BioMerieux, Marcy L'Etoile, France)) and incubated at 37˚C in 5% $CO_2$ for up to seven days. Bacterial cultures were identified with the Vitek 2 identification card (2009–2013) and the Vitek-MS system (2013–2015) (BioMérieux). The antibiotic susceptibilities of *S. agalactiae* isolates to amoxicillin, penicillin G, tetracycline, erythromycin, clindamycin, rifampicin, cotrimoxazole, vancomycin and teicoplanin were determined by agar disk diffusion method and interpreted according to the recommendations of the French Society for Microbiology and the European Committee on Antimicrobial Susceptibility Testing 2017 (http://www.sfm-microbiologie.org).

### Statistical analysis

Results are expressed as n (%) for categorical variables and median (minimum; maximum or Interquartile Range (IQR)) or mean (standard deviation (SD)) for continuous variables. Comparisons were done using the Fisher's exact tests for categorical variables. A p-value <0.05 was considered statistically significant. Data analyses were performed using GraphPad Prism version 6.00 for Mac OS X, (GraphPad Software, La Jolla California USA).

### Ethics approval and consent to participate

Approval from the Institutional review board of our university hospital was obtained (IRB Centre Hospitalier Universitaire Régional Caremeau, Nîmes, France N˚15/06.07). All procedures performed in the study were in accordance with the ethical standards and with the 1964 Helsinki Declaration and its later amendments or comparable ethical standards. Written informed consents have been obtained from all the patients included. No identification data are disclosed.

## Results

### Sociodemographic (Table 1)

Overall, 26 patients with *S. agalactiae* BJIs were identified. Twenty patients (77%) were male, median age was 62 years (range 27–80 years) and mean Charlson comorbidity index was 4.9 (SD 3.2). Most patients (22/26, 85%) had one or more underlying conditions. Diabetes mellitus was the most common comorbidity (14, 54%). The most common risk factors were tobacco use (10, 38%) followed by alcohol abuse (5, 19%). Four patients had solid cancer (15%) among whom three had radiotherapy on a bone that was infected afterwards. None of the patients was pregnant.

### Clinical characteristics (Table 1)

Only four patients (15%) had fever above 38.5˚C. Local erythema (17, 65%), pain (14, 54%) and purulent discharge (13, 50%) were the most frequent clinical signs. Bacteraemia was noted in four cases (15%) and septic shock in three cases (12%). Biological results showed that most patients had a biological inflammatory syndrome with elevated white blood cells count (16, 62%) and an elevated C-reactive protein (18, 69%) with a median of 155 mg/L (3.6–311 mg/L).

Overall, 15 (58%) patients had native BJI (11 osteomyelitis and four native septic arthritis), six (23%) patients had a prosthetic joint infection (five hip and one knee) and five (19%) patients had an internal fixation device infection (two in the ankle, two in the foot and one in

**Table 1. Demographic and clinical characteristics of the 26 patients with *S. agalactiae* bone and joint infection.**

| Characteristics | n = 26 |
|---|---|
| **Sex, male, [n(%)]** | 20 (77) |
| **Median age [years (range)]** | 62 (27–80) |
| **Comorbidities [n(%)]*** | 85 (22/26) |
| Alcohol abuse | 5 (19) |
| Tobacco use | 10 (38) |
| Cardiac failure | 2 (8) |
| Chronic liver disease | 3 (12) |
| Diabetes mellitus | 14 (54) |
| Inflammatory rheumatism | 1 (4) |
| Pregnancy | 0 (0) |
| Obesity** | 4 (15) |
| Immunodeficiency | |
| HIV | 1 (4) |
| Solid cancer | 4 (15) |
| Radiotherapy on infected bone | 3 (12) |
| Peripheral Arterial Obstructive Disease | 7 (27) |
| Peripheral neuropathy | 8 (31) |
| Diabetic foot | 8 (31) |
| **Charlson comorbidity Index, [mean (Standard deviation)]** | 4.9 (0–10) |
| **Clinical features [n(%)]** | |
| Fever (>38˚5) | 4 (15) |
| Purulent discharge | 13 (50) |
| Erythema | 17 (65) |
| Pain | 14 (54) |
| Bacteraemia | 4 (15) |
| Septic shock | 3 (12) |
| **Biological results** | |
| C-reactive Protein, (mg/L) [median (IQR)] | 155 (3.6–311) |
| White blood cells count > 10 G/L [n(%)] | 16 (62) |
| White blood cells, (G/L) [median (IQR)] | 17 (4.5–24.1) |

*Some patients had more than one comorbidity or risk factor.

**Obesity was defined as Body Mass Index > 30 kg/m$^2$.

the tibia). All patients with a prosthetic joint or internal fixation device had a delayed or late infection. The mean time between orthopaedic device implantation and infection onset was 137 months (range 4–300). Infections occurred between 4 months to 8 years after implantation for prosthetic joint infections, and between 4 months to 6 years for internal fixation devices.

Native BJIs were located to shoulder, sacrum (pressure ulcer) or sternum joint (n = 1), tibia or knee (n = 2), the majority were diabetic foot infections (8, 31%).

## Microbiological characteristics

Almost two-thirds of GBS BJIs were polymicrobial (16, 62%). The most common co-infecting agent was *Staphylococcus* sp. (9/16, 56%) with *S. aureus* (7/16) and *S. lugdunensis* (2/16), and *Enterococcus faecalis* (3/16, 19%).

Ten of the polymicrobial infections (10/16, 63%) were foot osteomyelitis: two patients with an internal fixation device infection and eight patients with diabetic foot infection.

Polymicrobial infections were more frequent in foot infections than in other locations (90% vs 44%, p = 0.0184). Six polymicrobial infections occurred in patients with prosthetic joint or internal fixation devices. There was no difference in polymicrobial infections rate between native (6/11, 54%) and prosthetic joint or internal fixation device infections (10/15, 67%) (p = 0.53).

All *S. agalactiae* isolates were susceptible to amoxicillin, benzylpenicillin, vancomycin and teicoplanin. Sixty percent of isolates were resistant to tetracycline, 35% to erythromycin and 20% to clindamycin. Only 14 isolates were tested for rifampicin: 8 (57%) were susceptible, four (29%) intermediate and two were resistant (14%). Rifampicin resistance was associated with cotrimoxazole resistance in both isolates, in addition to erythromycin resistance in one case and tetracycline resistance in the other.

In the three patients who died, one *S. agalactiae* strain was susceptible to all the antibiotics tested, one strain was only resistant to tetracycline and one strain was resistant to tetracycline, erythromycin and intermediate to rifampicin.

## Management (Tables 2 and 3)

Surgical treatment was performed in all six prosthetic joint infections: four had two-stage re-implantation (one died before the second prosthetic implantation); one had a one-stage re-implantation; one had conservative treatment (debridement only). All internal fixation devices were removed surgically with bone debridement. Half of the native bone/joint infections (7/ 15, 47%) had a surgical debridement.

All but two patients with prosthetic joint infections underwent probabilistic antibiotic regimen via a combination of vancomycin with a beta-lactam (ceftriaxone, cefotaxime or imipenem) and/or aminoglycoside (gentamicin). One patient had an initial ofloxacin monotherapy for a sepsis of unknown origin but switched to a combination of amoxicillin and gentamycin when *S. agalactiae* bacteraemia was diagnosed at day 4. The second patient had linezolid initial monotherapy having recently been treated with this antibiotic for a haematoma infected by *E. faecalis* and *S. aureus* in front of the prosthetic joint.

In patients with internal fixation device infections, two had empiric treatment of intravenous vancomycin (monotherapy or combined with ceftriaxone) and the other three patients

**Table 2. Medical management and outcomes of the 10 monomicrobial *S. agalactiae* bone and joint infections.**

| | Localisation | Initial antibiotic therapy (route of administration) | Duration | Final antibiotic therapy (route of administration) | Duration | Outcome |
|---|---|---|---|---|---|---|
| **Prosthetic joint infections** | Hip | Ofloxacin (oral)* | 4 days | Amoxicillin + Gentamycin (IV) | 4 days | Death |
| | Hip | Vancomycin + Cefotaxime* (IV) | 2 days | Amoxicillin + Rifampicin (oral) | 12 weeks | Reinfection |
| | Hip | Vancomycin + Gentamycin *(IV) | 6 days | Vancomycin (IV) | 12 weeks | Reinfection |
| | Hip | Linezolid (IV)* | 1 day | no | - | Death |
| **Internal fixation device infection** | Ankle | Amoxicillin/Clav+ Clindamycin (oral) | 4 weeks | no | - | Remission |
| **Native bone/joint infection** | Foot | Amoxicillin/Clav (IV)* | 2 weeks | Clindamycin + Rifampicin (oral) | 12 weeks | Remission |
| | Sternum | Amoxicillin + Gentamycin* (IV) | 12 days | Amoxicillin + clindamycin (oral) | 8 weeks | Remission |
| | Tibia | Pristinamycin (oral) | 6 weeks | no | - | Remission |
| | Knee | Amoxicillin + Gentamycin* (IV) | 10 days | Amoxicillin (oral) | 3 weeks | Death |
| | Shoulder | Amoxicillin + Gentamycin* (IV) | 7 days | Clindamycin (oral) | 5 weeks | Remission |

IV: Intraveinous, clav: Clavulanic acid

* probabilistic treatment.

**Table 3. Medical management and outcomes of the16 polymicrobial infections associated with *S. agalactiae* BJI.**

| | Localisation | Initial antibiotic therapy (route of administration) | Duration | Definite antibiotic therapy (route of administration) | Duration | Outcomes |
|---|---|---|---|---|---|---|
| **Prosthetic joint infections** | Knee | Vancomycin + ceftriaxone + gentamycin (IV)* | 4 days | Cotrimoxazole + Rifampicin (oral) | 12 weeks | Remission |
| | Hip | Vancomycin + Imipenem (IV)* | 4 days | Clindamycin + Fusidic acid (oral) | 12 weeks | Lost to follow-up |
| **Internal fixation device infection** | Ankle | Rifampicin + pristinamycin (oral) | 6 weeks | No | - | Remission |
| | Foot | Amoxicillin (IV)* | 7 days | Levofloxacin + rifampicin (oral) | 6 weeks | Remission |
| | Foot | Vancomycin + ceftriaxone (IV)* | 2 days | Clindamycin + rifampicin (oral) | 6 weeks | Remission |
| | Tibia | Vancomycin (IV)* | 1 day | Clindamycin (oral) | 6 weeks | Remission |
| **Native bone/joint infections** | Foot | AMC + ofloxacin (oral) | 18 days | no | - | Relapse |
| | Knee | Vancomycin + Gentamycin (IV)* | 3 days | Amoxicillin + rifampicin (oral) | 4 weeks | Remission |
| | Sacrum | Linezolid + Cotrimoxazole (oral) | 4 weeks | No | - | Remission |
| | Foot | AMC + Ofloxacin (oral | 11 weeks | No | - | Remission |
| | Foot | AMC + Ofloxacin (oral) | 6 weeks | No | - | Relapse |
| | Tibia | Amox/Clav (oral)* | 4 days | Rifampicin + levofloxacin (oral) | 6 weeks | Remission |
| | Foot | Amoxicillin + Cotrimoxazole (oral) | 5 weeks | No | - | Relapse |
| | Foot | PIP/Tazobactam + Vancomycin (oral)* | 6 days | AMC + Ofloxacin (oral) | 3 weeks | Remission |
| | Foot | PIP/Tazobactam (IV)* | 3 weeks | Ceftriaxone + Metronidazole (IV) | 3 weeks | Relapse |
| | Foot | AMC + Ofloxacin (oral) | 6 weeks | no | - | Relapse |

IV: Intravenous, AMC: Amoxicillin + clavulanic acid, PIP: Piperacillin.

* probabilistic treatment.

received adapted oral therapy. Among the 15 patients with native bone/joint infections, adapted antibiotic treatment was given to seven patients and an initial empiric treatment was given to the other eight patients of intravenous vancomycin combined with gentamicin or piperacillin/tazobactam, amoxicillin combined with gentamicin, piperacillin/tazobactam in monotherapy or amoxicillin/clavulanic acid in monotherapy.

## Clinical outcomes (Tables 2 and 3)

One patient was lost to follow-up. Follow-up was complete for the 25 remaining patients. Among them, three died (3/25, 12%), 15 were in remission (15/25, 60%), and seven had treatment failure (7/25, 28%) of whom five had polymicrobial infection on a diabetic foot. Treatment failure were relapse in five patients (including polymicrobial infection with *S. agalactiae* in two cases) and reinfection in two cases. The latter two patients had hip prosthesis infection, both whom had had their prosthesis removed via a 2-stage revision strategy. In the first patient, *M. morganii* and *S. epidermidis* were identified during the second stage re-implantation. The other patient relapsed eight months after the second re-prosthesis implantation from *E. faecalis* infection.

Neither diabetes mellitus (p = 0.13) nor concomitant bacteraemia at the time of diagnosis (p = 0.28) were statistically associated with treatment failure.

All three patients who died had a high Charlson score (from 8 to 11) and a monomicrobial *S. agalactiae* infection. One had a native knee infection and two had a hip prosthesis infection. Both dead when their prosthetic joint were removed.

## Discussion

*S. agalactiae* is a growing cause of invasive infections in adults. We report 26 *S. agalactiae* BJIs from January 2009 to June 2015. Most of the infections occurred in patients with

comorbidities. Diabetes mellitus is a known major risk factor for GBS BJIs, present in 20 to 48% of patients [6,10–12] and associated with unfavourable clinical outcome [10].

Two-thirds of the patients had polymicrobial infection (mainly with *S. aureus* and coagu-lase-negative staphylococci) which differs from the series of Seng *et al.* including 37 cases of BJIs due to *S. agalactiae* that were usually monomicrobial infections [10]. The rate of bacterae-mia did not differ between monomicrobial and polymicrobial infections and was similar to that reported in the series of Seng (15%) [10]. In contrast to results found by Fiaux *et al.* [13], concomitant bacteraemia at the time of diagnosis did not negatively impact the outcome for streptococcal prosthetic infections.

The management of GBS BJIs was very heterogeneous in our patients depending on the time until infection, the presence of orthopaedical device or multiple bacteria. Fiaux *et al.* sug-gested that rifampicin combined with another agent, especially levofloxacin, was an effective and well-tolerated treatment in their series of 95 streptococcal prosthetic joint infections [13]. In our study, among monomicrobial *S. agalactiae* BJI, two combinations with rifampicin were used, once with a good outcome and the second with a relapse with other bacteria. In polymi-crobial BJI with *S. agalactiae* and *S. aureus*, four oral combined antibiotic treatments with rifampicin and a second agent were given with a good outcome. Previous studies have shown that rifampicin was active against the majority of *Streptococcus* strains [10,13]. In our experi-ence, only 57% of tested strains were susceptible to this antibiotic. This result suggests that rifampicin should be tested in order to preferentially propose this antibiotic in the GBS pros-thetic BJIs therapeutic regimen.

The rate of relapse observed in our study was high (27%), in accordance with rates reported in the literature (18–32.7%) [10,13,14] which is probably due to the fact that 5 out of 7 relapses occurred in patients with diabetic foot infection and a polymicrobial infection. Our mortality rate was high (12%) but comparable to other studies [14,15]. Interestingly, the three patients who died with a *S. agalactiae* BJI in our study had a Charlson score > 8. *S. agalactiae* related BJIs are known to be more severe with a poorer outcome as showed by Zeller *et al*. In 2009, this team described 24 cases of *S. agalactiae* prosthetic hip infections, compared to 115 other-pathogens cases. Their findings showed that *S. agalactiae* prosthetic hip infection had a poorer outcome compared to other-pathogens [14].

This work had some limitations. First, this was a monocentric study in mainland France, with an observational retrospective design and a small sample size. Second, the studied popula-tion was heterogeneous with monomicrobial and polymicrobial infections, native and pros-thetic joints infections, including diabetic foot infections, which makes management recommendations difficult.

## Conclusion

We observed an heterogenous management and a high rate of relapse in our patients with *S. agalactiae* BJIs probably due to the fact that infections were often polymicrobial diabetic foot infection.

## Supporting information

**S1 Dataset.**
(XLSX)

## Acknowledgments

We would like to thank Sarah Kabani for editing the manuscript.

## Author Contributions

**Conceptualization:** Paul Loubet.

**Data curation:** Yatrika Koumar.

**Formal analysis:** Paul Loubet, Yatrika Koumar.

**Investigation:** Sophie Schuldiner.

**Methodology:** Nicolas Cellier, Albert Sotto.

**Supervision:** Nicolas Cellier, Jean-Philippe Lavigne, Albert Sotto.

**Validation:** Catherine Lechiche, Sophie Schuldiner, Pascal Kouyoumdjian, Albert Sotto.

**Writing – original draft:** Paul Loubet, Yatrika Koumar.

**Writing – review & editing:** Sophie Schuldiner, Pascal Kouyoumdjian, Jean-Philippe Lavigne, Albert Sotto.

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
