## [Decision Letter · Decision Letter 0]

4 Jan 2021

PONE-D-20-36123

Clinical features and outcome of Streptococcus agalactiae bone and joint infections over a 6-year period in a French University Hospital

PLOS ONE

Dear Dr. Loubet,

Thank you for submitting your manuscript to PLOS ONE. After careful consideration, we feel that it has merit but does not fully meet PLOS ONE’s publication criteria as it currently stands. Therefore, we invite you to submit a revised version of the manuscript that addresses the points raised during the review process.

I would like to thank you for your efforts; however, as raised by reviewer 1, there are major concerns about your work.

Kindly respond to all comments carefully and show what your findings will be adding to the current literature.

Without a clear response to all comments, I am afraid that the manuscript cannot move forward.

We look forward to receiving your revised manuscript.

Kind regards,

Sherief Ghozy, M.D., Ph.D. candidate

Academic Editor

PLOS ONE

Journal Requirements:

2. Thank you for including your ethics statement:  "Approval from the Institutional review board of our university hospital was obtained (IRB N°15/06.07). All procedures performed in the study were in accordance with the ethical standards and with the 1964 Helsinki Declaration and its later amendments or comparable ethical standards. Informed consents have been obtained. No identification data are disclosed.".   

Please amend your current ethics statement to include the full name of the ethics committee/institutional review board(s) that approved your specific study. Once you have amended this/these statement(s) in the Methods section of the manuscript, please add the same text to the “Ethics Statement” field of the submission form (via “Edit Submission”).

3. Please provide additional details regarding participant consent. In the ethics statement in the Methods and online submission information, please ensure that you have specified what type you obtained (for instance, written or verbal, and if verbal, how it was documented and witnessed). If your study included minors, state whether you obtained consent from parents or guardians.

4. Please include the date(s) on which you accessed the databases or records to obtain the data used in your study.

5. Please report any exclusion criteria used to select patients in your study.

6. Thank you for stating the following financial disclosure:

"The funders had no role in study design, data collection and analysis, decision to publish, or preparation of the manuscript"

7. Your ethics statement should only appear in the Methods section of your manuscript. If your ethics statement is written in any section besides the Methods, please move it to the Methods section and delete it from any other section. Please ensure that your ethics statement is included in your manuscript, as the ethics statement entered into the online submission form will not be published alongside your manuscript.

Reviewers' comments:

Reviewer's Responses to Questions

**Comments to the Author**

1. Is the manuscript technically sound, and do the data support the conclusions?

Reviewer #1: Yes

Reviewer #2: Yes

2. Has the statistical analysis been performed appropriately and rigorously? 

Reviewer #1: Yes

Reviewer #2: Yes

3. Have the authors made all data underlying the findings in their manuscript fully available?

Reviewer #1: Yes

Reviewer #2: Yes

4. Is the manuscript presented in an intelligible fashion and written in standard English?

Reviewer #1: Yes

Reviewer #2: Yes

5. Review Comments to the Author

Reviewer #1: 1. This manuscript is a retrospectively identified case series of 26 patients (62% with polymicrobial infection) with Streptococcus agalactiae bone and joint infections (BJIs), describing their clinical features and treatment outcomes from a single center.

2. The manuscript lacks of any novel information regarding epidemiology, clinical features, or treatment outcomes of bone and joint infections caused by S. agalactiae.

3. The conclusions in the abstract are not supported by data presented.

4. "S. agalactiae BJIs mainly affected diabetic patients" - In a case series with no comparison group, this statement is unwarranted. A more appropriate statement would be "Diabetes mellitus was the most common comorbidity among these patients."

5. "Treatment failure was frequent, probably due to the complexity of BJIs requiring a multidisciplinary management." There is no statistical analysis presented to support the statement that complexity of BJIs requiring multidisciplinary management was associated with treatment failure.

6. "The most common risk factors were tobacco use (10, 38%) followed by alcohol abuse (5, 19%)." Risk factors for what?

Reviewer #2: We had pleasure to read the article which is rich of informations that help to better understand the BJIs in the context of a french teaching Hospital. The methodology was rigorously performed and results are clearly shown and verifiable. In order to increase more the value of the article, we had some suggestions that doesn’t affect substantively the study but just the shape.

1- The first time the acronym PTI appear in the article, it doesn’t follow by the signification. So it will be better to explain so early the explanation of the abbreviation.

2- The rate of deaths appear in the article to be 2/25 or the actual patients included in the study is 26. The explanation of why you used 25 rather than 26 appear at line 173 with the one patient lost of follow up.

The authors should look how to early inform this in the results of the study.

6. PLOS authors have the option to publish the peer review history of their article (what does this mean?). If published, this will include your full peer review and any attached files.

Reviewer #1: No

Reviewer #2: **Yes: **Franck David ABOUNA

---

## [Author Response · Author response to Decision Letter 0]

17 Feb 2021

Editor

 Authors’ response: We have checked these points.

2. Thank you for including your ethics statement: "Approval from the Institutional review board of our university hospital was obtained (IRB N°15/06.07). All procedures performed in the study were in accordance with the ethical standards and with the 1964 Helsinki Declaration and its later amendments or comparable ethical standards. Informed consents have been obtained. No identification data are disclosed.". 

Please amend your current ethics statement to include the full name of the ethics committee/institutional review board(s) that approved your specific study. Once you have amended this/these statement(s) in the Methods section of the manuscript, please add the same text to the “Ethics Statement” field of the submission form (via “Edit Submission”).

 Authors’ response: As requested, the full name of the IRB has been added and the ethic statement moved to the materials and methods section.

3. Please provide additional details regarding participant consent. In the ethics statement in the Methods and online submission information, please ensure that you have specified what type you obtained (for instance, written or verbal, and if verbal, how it was documented and witnessed). If your study included minors, state whether you obtained consent from parents or guardians.

 Authors’ response: As requested, we have specified the type of consent we obtained

4. Please include the date(s) on which you accessed the databases or records to obtain the data used in your study.

 Authors’ response: As requested, dates on which we accessed the records to obtain the data were added in the materiel and methods section.

5. Please report any exclusion criteria used to select patients in your study.

 Authors’ response: As requested, exclusion criteria has been added. 

Page 3, line 71 : Those with non-bone and joint infection were excluded.

6. Thank you for stating the following financial disclosure:

"The funders had no role in study design, data collection and analysis, decision to publish, or preparation of the manuscript"

At this time, please address the following queries :

a. Please clarify the sources of funding (financial or material support) for your study. List the grants or organizations that supported your study, including funding received from your institution.

d. If you did not receive any funding for this study, please state: “The authors received no specific funding for this work.”

 Authors’ response: As requested, we have clarified the financial disclosure. 

7. Your ethics statement should only appear in the Methods section of your manuscript. If your ethics statement is written in any section besides the Methods, please move it to the Methods section and delete it from any other section. Please ensure that your ethics statement is included in your manuscript, as the ethics statement entered into the online submission form will not be published alongside your manuscript.

 Authors’ response: As requested this change has been done.

Reviewer #1: 

This manuscript is a retrospectively identified case series of 26 patients (62% with polymicrobial infection) with Streptococcus agalactiae bone and joint infections (BJIs), describing their clinical features and treatment outcomes from a single center.

2. The manuscript lacks of any novel information regarding epidemiology, clinical features, or treatment outcomes of bone and joint infections caused by S. agalactiae.

Authors’ response: The literature on bone and joint infections specifically due to GBS is scarce. To the best of our knowledge, the largest series to date is the one from Kerneis et al (Ref 11 in our article) published in 2017 including 163 patients. In this paper there are only patients’ description and data on resistances of the isolated strains of S. agalactiae. In the article from Seng et al. (Ref 10 in our article) presenting clinical features and outcomes of bone and joint infection with Streptococcus involvement there is no specific data on the treatment used and clinical outcomes of the 37 GBS bone and joint infections included. Conversely to these two works, our manuscript displays specific information on antibiotics treatment used and clinical outcomes in mono (Table 2) and polymicrobial (Table 3) GBS BJI patients.

3. The conclusions in the abstract are not supported by data presented.

Authors’ response: We have made the following changes in the conclusion of the abstract (see also comment 4 and 5)

Diabetes mellitus was the most common comorbidity. We observed an heterogenous management and a high rate of relapse.

4. "S. agalactiae BJIs mainly affected diabetic patients" - In a case series with no comparison group, this statement is unwarranted. A more appropriate statement would be "Diabetes mellitus was the most common comorbidity among these patients."

Authors’ response: We have made the change suggested by the reviewer.

5. "Treatment failure was frequent, probably due to the complexity of BJIs requiring a multidisciplinary management." There is no statistical analysis presented to support the statement that complexity of BJIs requiring multidisciplinary management was associated with treatment failure.

Authors’ response: As suggested by the reviewer we have changed this sentence by the following : “We observed an heterogenous management and a high rate of relapse.”

6. "The most common risk factors were tobacco use (10, 38%) followed by alcohol abuse (5, 19%)." Risk factors for what?

Authors’ response: We agree with the reviewer that this sentence is not very clear and is furthermore not relevant for the topic . We have decided to remove this sentence from the text and the term “risk factor” from Table 1.

Reviewer #2: 

We had pleasure to read the article which is rich of informations that help to better understand the BJIs in the context of a french teaching Hospital. The methodology was rigorously performed and results are clearly shown and verifiable. In order to increase more the value of the article, we had some suggestions that doesn’t affect substantively the study but just the shape.

1- The first time the acronym PTI appear in the article, it doesn’t follow by the signification. So it will be better to explain so early the explanation of the abbreviation.

Authors’ response: As requested by the reviewer, we have added the meaning of PJI in the abstract.

2- The rate of deaths appear in the article to be 2/25 or the actual patients included in the study is 26. The explanation of why you used 25 rather than 26 appear at line 173 with the one patient lost of follow up.

The authors should look how to early inform this in the results of the study.

Authors’ response: The total of 25 is only used for clinical outcomes and this is where we chose to explain that one patient was lost to follow-up. We think it might be confusing to explain the lost to follow-up earlier in the results section as the first parts of the results are displayed on 26 patients.

---

## [Editor Report · Decision Letter 1]

23 Feb 2021

Clinical features and outcome of Streptococcus agalactiae bone and joint infections over a 6-year period in a French University Hospital

PONE-D-20-36123R1

Dear Dr. Loubet,

We’re pleased to inform you that your manuscript has been judged scientifically suitable for publication and will be formally accepted for publication once it meets all outstanding technical requirements.

Kind regards,

Sherief Ghozy, M.D., Ph.D. candidate

Academic Editor

PLOS ONE

---

## [Editor Report · Acceptance letter]

4 Mar 2021

PONE-D-20-36123R1 

Clinical features and outcome of *Streptococcus agalactiae* bone and joint infections over a 6-year period in a French University Hospital 

Dear Dr. Loubet:

I'm pleased to inform you that your manuscript has been deemed suitable for publication in PLOS ONE. Congratulations! Your manuscript is now with our production department. 

Kind regards, 

on behalf of

Dr. Sherief Ghozy 

Academic Editor

PLOS ONE